# A Structural Proteomics Exploration of Synphilin-1 and Alpha-Synuclein Interaction in Pathogenesis of Parkinson’s Disease

**DOI:** 10.3390/biom14121588

**Published:** 2024-12-12

**Authors:** Asmita Tripathi, Rajkrishna Mondal, Malay Mandal, Tapobrata Lahiri, Manoj Kumar Pal

**Affiliations:** 1Department of Applied Sciences, Indian Institute of Information Technology Allahabad, Prayagraj 211015, India; rss2019009@iiita.ac.in; 2Department of Biotechnology, Nagaland University, Kohima 797004, India; rajkrishna@nagalanduniversity.ac.in; 3Department of Medicine, Section of Rheumatology and Gwen Knapp Center for Lupus and Immunology Research, University of Chicago, Chicago, IL 60637, USA; 4Faculty of Engineering & Technology, United University Prayagraj, Prayagraj 211012, India; manoj.pal@uniteduniversity.edu.in

**Keywords:** Parkinson’s disease, Synphilin-1 and alpha-synuclein interaction, protein–protein interaction, protein aggregates, Lewy bodies, prediction of large protein structures

## Abstract

Pathological significance of interaction of Synphilin-1 with mutated alpha-synuclein is well known to have serious consequences in causing the formation of inclusion bodies that are linked to Parkinson’s disease (PD). Information extracted so far pointed out that specific mutations, A53T, A30P, and E46K, in alpha-synuclein promote such interactions. However, a detailed structural study of this interaction is pending due to the unavailability of the complete structures of the large protein Synphilin-1 of chain length 919 residues and the mutated alpha-synuclein having all the reported specific mutations so far. In this study, a semi-automatic pipeline-based meta-predictor, AlphaLarge, is created to predict high-fidelity structures of large proteins like Synphilin-1 given the limitations of the existing protocols. AlphaLarge recruits a novel augmented AlphaFold model that uses a divide and conquer based strategy on the foundation of a self-sourced template dataset to choose the best structure model through their standard validations. The structure models were re-validated by a Protein Mediated Interaction Analysis (PMIA) formalism that uses the existing structurally relevant information of these proteins. For the training dataset, the new method, AlphaLarge, performed reasonably better than AlphaFold. Also, the new residue- and domain-based structural details of interactions of resultant best structure models of Synphilin-1 and both wild and mutated alpha-synuclein are extracted using PMIA. This result paves the way for better screening of target specific drugs to control the progression of PD, in particular, and research on any kind of pathophysiology involving large proteins of unknown structures, in general.

## 1. Introduction

It is well known that the substantially below normal level of dopamine is linked to cognitive impairment as well as abnormal motor activities causing tremors and rigidity [1]. Consequently, the pathological manifestation of Parkinson’s disease (PD) mainly happens due to the progressive degradation of the dopaminergic neurons that is mainly represented through the manifestation of Lewy Bodies (LB), located in the brains of PD patients as fibrillar aggregates [2]. The complexity of the pathology related to PD is yet to be fully explored. However, current research indicates that there are three main triggering points to cause LB aggregates within brain. Among them is the report on the mutation in the SNCA gene that codes for the protein, alpha-synuclein (a-Syn), also located in the presynaptic neurons [3]. It is known that overexpression of a-Syn has a toxic effect and causes neuronal death through their aggregation [4]. The second one is the presence of another protein, Synphilin-1 (Syn-1), within LB aggregates that appears to be another suspect in causing LB aggregates [5]. Overexpression of Syn-1 is also reported to render a similar toxic effect to cause LB aggregates like the mutated a-Syn [6]. The third interesting event is the interaction of these two proteins, mutated a-Syn and Syn-1, both of which are found within LB aggregates [7]. In this regard, the role and function of Syn-1 in the pathogenesis of PD need to be studied to its logical conclusion. The experimental structure prediction method, NMR, provides partial information for structure of Synphilin-1 [8]. Elucidating the complete structure of Synphilin-1 will enhance understanding of its role in Parkinson’s disease. Although Alphafold2 [9] has predicted the structure of Synphilin-1 as deposited in the UniProt database, there appears to be sufficient scope for improvement in the available predicted structure of the target protein. Also, the first part of our study shows that the Alphafold2-modeled structure fails to show high binding affinity with the most appropriate ligands.

Among the many causes responsible for the pathological manifestation of PD, we have focused our study to examine the effect of the simultaneous occurrence of a specific combination of mutations, A53T, A30P, and E46K in a-Syn, and its effect in the interaction of Syn-1 with mutated a-Syn leading to the formation of Lewy Bodies (LB)—an important cause of PD. As discussed above, the primary concern of this work is to explore the detailed structural aspects of interaction of Syn-1 with mutated a-Syn to find the specific binding sites and, subsequently, to help in finding preventive therapeutics through design of target specific drug. In this regard, the main bottlenecks in this study are found to be the unavailability of the experimentally determined structures of the large protein, Syn-1, which is of size 919 residues and the a-Syn having all the reported mutations, e.g., A53T, A30P, and E46K [10,11,12]. For any structural studies of a protein–protein interaction, the availability of highly authentic structures is considered to be a good starting point, whereas, acceptability of the predicted structures by earlier structure predictor models was questionable before the advent of the Google Colab [13] designed AlphaFold, which showed remarkably good performance leading to further use of structural models predicted by it. Alphafold2 almost exactly solves one of the major problems i.e., protein three-dimensional structure prediction of whole human proteome, and, thus, gained more prominence than other structure–prediction methods [14]. However, it is observed that, like other earlier predictors, AlphaFold2 also suffers from some limitations, especially in predicting structures of large proteins [15,16,17,18]. As reported by Lee et al., 2022 [19], the major limitation of deep learning-based methods like AlphaFold and RoseTTAFold is in predicting the intricate multi-domain large protein structure, primarily the domain interacting region. They also pointed out that the performance of AlphaFold is highly dependent on the quality and diversity of the training data. Therefore, unavailability of a significant number of high-quality structural data for large size protein structure in the training dataset may lead to the inaccuracies in the predicted structure. Additionally, the prediction accuracy of AlphaFold2 can be compromised for proteins with shallow-depth multiple sequence alignments (MSA), i.e., those with fewer homologous sequences or low sequence similarity [9]. Therefore, in this work, a semi-automatic pipeline is proposed to design a meta-predictor, AlphaLarge, that is basically an augmented version of AlphaFold. AlphaLarge works on a divide and conquer strategy in which the structures of overlapping small fragments of target sequence are first predicted by AlphaFold2 for their reuse as templates within the Homology Model [20] application to substantially enhance prediction accuracy in comparison to the direct application of AlphaFold2.

Studies on protein–protein and protein–ligand interactions are always of considerable interest because of their potential in revealing pathophysiological mechanism of a disease process [21,22,23,24]. In this work, information on these interactions are extracted to fulfill two different objectives. First off, the already known information about the interaction of a ligand or protein with the target protein is utilized to select the best structure models from the output generated by the semiautomatic meta-predictor, AlphaLarge. The second objective is to study the interaction of Syn-1 with mutated a-Syn to know about the overall binding characteristics, including the binding site residues of a-Syn for controlling this interaction-mediated toxicity leading to PD. Therefore, a Protein Mediated Interaction Analysis (PMIA) Paradigm is introduced in this work to consistently bring out the crucial binding characteristics of such interactions. Under the framework of PMIA, an interaction is characterized by its binding energy, binding site residues participating in this interaction (for protein–protein interaction), or binding site residues of the target protein bonding with the ligands (for protein–ligand interaction) and, the bonds, formation of which causes this interaction. Focusing on the main objective of study on the structural details of the Syn-1 and a-Syn interaction, this work emphasized the development of an analytical formalism for this study combining the potential of AlphaLarge with PMIA. The new and specific binding site information extracted through this work is found to be consistent with the experimental reports with the potential to design better therapeutic strategy against the progress of PD.

## 2. Materials and Methods

This work supports building an analytical formalism to study the specific structural details of the interaction between Syn-1 and mutated a-syn. Following this workflow, this section is divided into three sub-sections. The first sub-section deals with the formalism developed to analyze the specific structural details of the binding sites of Syn-1 and a-Syn participating in their interaction considering the availability of their respective structures. The features, e.g., binding energy, participating residues or atoms, and the interaction bonds are studied following specific rules under the framework PMIA as described in the Introduction. The second subsection deals with the building of the semi-automatic pipeline to create the meta-predictor, AlphaLarge to get highly accurate structures required for this study.

### 2.1. Protein Mediated Interaction Analysis (PMIA) Formalism

Once considerably accurate structure models of the target protein (TP) and its interacting partner (IP) (another protein or a ligand) are available, the following rule is used to extract specific information on their binding for two purposes. The first one is to select the best structure model output by the AlphaLarge and the second one is to extract specific structural details of binding for the interaction between Syn-1 and mutated a-syn.

Rule 1: Calculate the binding energy of interaction. In this work, it is calculated using the module Maestro of the application Schrodinger. Select the structure model that exhibits the least binding energy of interaction.

Rule 2: Re-validate the structure selected through rule 1 by first finding out the binding site residues or domains along with specific interaction information of the target protein. These interacting residues or domains should correspond well with those already reported. Otherwise, the selected structure model is discarded for any further study. The parameters utilized in PMIA formalism are depicted in Figure 1.

### 2.2. Building of Semi-Automatic Pipeline for Designing the Meta-Predictor AlphaLarge

#### 2.2.1. Collection of Data for Sample Proteins and Ligands

Since this work aims to get the structure of the protein, Syn-1, a single chain protein of sequence length 919, the range of the sequence length for the protein samples collected in this work is also kept similar within the limits of 900 to 1100. To calculate a statistically acceptable sample size, *S*, the following analytical formula of Charan et al. (2013) [25] is utilized as shown in Equation (1):(1)$$S=\frac{Z^{2}\times p\left(1-p\right)}{d^{2}}=14.31$$
where, *p* is the fraction of protein following the above-mentioned criteria within the population and is obtained as *p* = n/N = 0.0094 for the population N = 72213 and, number of proteins following the above-mentioned criteria, n = 679, standard normal variate Z = 1.96 within a level of significance, d = 0.05.

In this work, after rounding off, the sample size was considered as S = 14 and utilized further for this work. Table 1 shows these sample proteins and their corresponding known ligands as collected from PDB.

#### 2.2.2. Working Principle of AlphaLarge

The semiautomatic pipeline-based meta-predictor, AlphaLarge, designed in this work to get structures of single-chain proteins, is based on the steps as given below. It should be noted that prior prediction of domain from the protein sequence by current standard methods (e.g., ThreaDom, Pfam) [49,50] is avoided in this work for the paradox that domains are also a large part of the structure only and their occasional incorrect prediction [51] may have a cascading effect in cumulative error accumulation.

Step 1: If the protein size is below 400, find the structure by the Direct Application of AlphaFold2 (referred as DAA pipeline), otherwise go to Step 2. Naturally, structure models generated using the DAA pipeline are referred as DAA_models.

Step 2: If the protein size is more than 400, then divide it into overlapping segments to restrict the size of each of the fragments as 400, keeping the length of the overlapping sections of consecutive sequence fragments as 100. It also corresponds well with Rost’s criteria [52], which says that the probability of finding the homologue for a protein decreases with the increase in the protein’s sequence length. Therefore, depending on the size of the protein, the last fragment size is compensated accordingly. For a hypothetical example, for a protein of size 885, the residue count for each of the fragments will be as follows:

First fragment: 1 to 400

Second fragment: 301 to 700

Third fragment (i.e., the last fragment): 601 to 885

For details of the algorithm, the MATLAB code for fragmentation is given in the Appendix A. The details on the residue counts for each of the fragments of all the training proteins and Syn-1 are given in Appendix A. Also, it is found that the small changes (with a change in the fragment size between 300 and 400 and overlapping size between 100 and 200) made in the fragmentation algorithm do not alter the structures of the predicted models used in this work.

Step 3: Predict the structures for all the fragments using AlphaFold2 Colab Notebook to make them qualify to be templates for the target protein. These fragments with their predicted structures are referred to as derived templates in this work. Figure 2 demonstrates the divide and conquer strategy used in the semiautomatic pipeline-based meta-predictor, AlphaLarge.

Step 4: The alignment of each one of the derived templates with the target protein was performed using Clustal Omega [53] of PyMod 3.0 plugin nested in PyMOL application.

Step 5: The structure of the target protein was modeled by homology modeling based plugin Modeller available in PyMOL application. Steps 2 to 5 are referred as the Derived Template Source (DTS) pipeline and the structure models generated using it is referred as DTS_models.

Step 6: Model protein structures are validated using standard validation parameters, ERRAT [54], Ramachandran score [55], G-factor [55], SRI [56], PAL–RMSD [57], and QMEAN-score [58]. For training sample proteins of known experimentally determined structures, GDT–TS [59], and TM-Score [60] are also computed.

Step 7: Model protein structures are re-validated using their binding characteristics with their corresponding ligands and matching them with the published reports following PMIA as described in Section 2.1. In this step, the following applications are used for their specific purposes:

Schrodinger–Maestro Release 2017: For the study of protein–ligand docking [61] and to calculate the binding energy (i.e., Glide’s XP docking scoring function) the following [62] are used:

PatchDock: For the study of protein–protein interaction [63]

PRODIGY: For the study of the binding energy of the interacting complex [64]

BIOVIA Discovery studio visualizer 4.5: For visualization of docking results [65]

MATLAB version R2022a: For computing proximity of binding site centre of experimental structure with that of DAA and DTS

Step 8: The best model structure is selected following the criteria depicted in rule 2 of PMIA formalism using steps 6 and 7.

A flowchart depicting the working principle of AlphaLarge is shown in Figure 3. The working principle of AlphaLarge is validated using the sample proteins given in Table 1. Subsequently, it is applied to get the structures of the proteins of the present case study, e.g., Syn-1 and a-Syn. Finally, the interaction between Syn-1 and a-Syn is studied under PMIA formalism to extract the structural details of their binding leading to their aggregates.

To get the structures of Syn-1 using AlphaLarge, first the sequence of Syn-1 was retrieved from UniProt database employing accession no. Q9Y6H5. Subsequently, this sequence is fed into AlphaLarge to get the best structure model of Syn-1. For the wildtype a-Syn, the experimentally obtained human micelle-bound protein with PDB ID 1XQ8 was used. In case of the mutated version of a-Syn, first, its sequence was retrieved from the UniProt using accession no. P37840. In the next step, this sequence is manually altered through the mutations, A30P, A53T, and E46K. Finally, the structure of a-Syn is obtained through the direct application of AlphaFold2.

## 3. Results

Considering the objective of this work to extract the structural mechanism of interaction between Syn-1 and a-Syn leading to toxic LB aggregate formation, the result presented in Section 3.1 shows the efficiency of the semiautomatic pipeline-based meta-predictor, AlphaLarge, to predict authentic structure models of the training protein samples for which the structures are known and therefore, validation is easier. After the potential of AlphaLarge is established, the result presented in Section 3.2 shows the standard validations of the structure models generated by AlphaLarge for both Syn-1 and mutated a-Syn with specific 3 mutations, which are not available currently by any experimental means. Finally, the result of the structural and functional analysis of interaction between Syn-1 and a-Syn is provided in the result Section 3.3.

### 3.1. Results of AlphaLarge Application on the Training Samples

The quality of DAA and DTS modeled structures under the AlphaLarge protocol was compared with their experimentally known structure employing standard structure validation parameters as given in step 6 of Section 2.2.2 to select the best structure model. The comparison of DTS with DAA was performed on the basis of the following two criteria for a specific validation parameter:

Win: $W_{DTS}$ and $W_{DAA}$ refers to situations where the validation parameter is found to be better for DTS and DAA structure models, respectively.

Tie: Tie refers to the situation where the absolute difference between the validation scores for DTS and DAA models are less than a predefined value, say for δ, which is fixed for a specific validation parameter as the 0.01% of its best standard value.

Table 2 clearly depicts the percentage of wins, losses, and tie cases for structure generated by DAA and DTS methods. It was found that DTS structures outperformed DAA structures in terms of standard validation parameters, Ramachandran score, G-factor, PAL_RMSD, and QMEAN whereas, DAA structures won in terms of ERRAT, GDT–TS, and TM-score. Also, as shown in Table 2 in the case of Ramachandran score, DTS structures completely win over DAA structures, which clearly indicate that the stereochemical factors of the modeled structures using derived templates are reasonably good. If the all atom RMSD (PAL_RMSD) score is taken into account, DTS structures clearly perform better than DAA structures. The PAL_RMSD score is a more reliable parameter for structure validation through comparison with experimentally obtained structure than usual RMSD score. Hence, both methods, DTS and DAA, perform competitively with each other with respect to standard validation parameters.

The best structure models are selected using the winning or tie performance of the sub-pipelines, DTS and DAA. Also, the results of PMIA formalism as shown in the latter section in Table 3 describing the corresponding bonds resolved as well as the binding energy, respectively, are used for the purpose of re-validating the structure models as well as to break the tie for final selection of a model.

Re-Validation of the DTS and DAA Structure Models Using PMIA Formalism

To re-validate the findings obtained through standard validation protocols, the interaction-based PMIA formalism was also applied for the structure models obtained through DAA and DTS pipelines. It is also useful to break the tie cases as mentioned above. In this regard, the result of the application of PMIA for the study of protein–ligand interactions for each of the sample proteins is presented in Table 3. It shows the performances of DTS and DAA modeled structures in comparison to that reported from the experiments.

While Table 3 shows that the performance of DTS is slightly better than DAA in resolving the total number of interactions, in individual cases, it appears to be quite different. It demonstrates the competitive performance of DTS and DAA sub-pipelines for individual proteins and the need of building the semiautomatic pipeline-based meta-predictor, AlphaLarge, based on the competitive performance of DTS and DAA. This finding is again found to correspond well with the comparative profile of binding energies of known interactions, BI_DTS_ and BI_DAA_, obtained for individual proteins using DTS and DAA approaches, respectively, as shown in Table 3 (details of the residues are given in Appendix A).

As shown in Table 3, the data for the binding energies of known interactions, BI_Experimental_, BI_DTS_, and BI_DAA_, obtained for individual sample proteins using experimental, DTS, and DAA models, respectively, also shows that the performance of DTS is reasonably better than DAA in terms of two parameters, “proximity of binding site centre of experimental structure with that of DAA and DTS models” and “proximity of binding energy of experimental structure with that of DAA and DTS models” (please see Table 1 for residue and domain details of binding sites). It also indicates the competitive performance of DTS and DAA for individual proteins and the need of building the semiautomatic pipeline-based meta-predictor, AlphaLarge, based on the competitive performance of DTS and DAA.

### 3.2. Standard Validation of Syn-1 Model Structures

After testing the DTS and DAA pipelines of AlphaLarge, it was pertinent to examine the potential of these methods for the pathologically important protein, synphilin-1, as a case study for which no experimental or predicted structure is available. Naturally, among the standard validation parameters used in the training phase of developing AlphaLarge, whichever was found dependent on the original structure was discarded for their utilization in this case study. Table 4 summarizes the result of validation of Synphilin-1 structures obtained from DAA and DTS using ERRAT, Ramachandran score, G-factor, QMEAN Score, and % coil amount. For non-bonded atomic interactions, the comprehensive quality factor is determined by ERRAT. The result shows that DTS predicted structure is better than that predicted by DAA.

As reported in Table 4, the Ramachandran plots generated for Synphilin-1 in Figure 4 also showed that the stereochemical property of DTS predicted structure is much better than that predicted by DAA. The detailed analysis of Table 4 reveals that most of the residues of DTS structure (82.1%) are within the highly allowed (i.e., core) region in comparison to the DAA structure (32.2%). Therefore, it is quite natural that an abundance of DTS residues in other regions (i.e., disallowed, generously allowed, and allowed) will be less than the DAA structure. This observation is found to be consistent with the Ramachandran plot shown in Figure 4. Similarly, DTS structure shows G factor value (−0.15) within an allowed limit of −0.5 to 0 whereas, for DAA (−2.55), the value is much further away from this limit. Regarding QMEAN Score, the DTS value (−5.97) is much closer to its expected score (i.e., 0) in comparison to DAA (−21.82). The same trend follows for %coil and ERRAT scores where DTS structure substantially outperformed DAA structure.

#### 3.2.1. Result of PMIA Formalism Based Re-Validation of Syn-1 Structure Using Binding Energy

Alpha-synuclein, an intrinsically disordered protein, undergoes structural transitions under specific conditions, including a notable shift from an alpha-helix to a beta-sheet conformation. In its interaction with lipid membranes (PDB ID: 1XQ8) and proteins like calmodulin (PDB ID: 2M55), alpha-synuclein adopts a stable alpha-helical structure. Therefore, for the study of this interaction, the experimentally obtained structure of a-Syn (wildtype Human micelle-bound alpha-helical conformation of full-length alpha-synuclein with PDB ID 1XQ8) was used. Finally, the analysis of binding energy of Synphilin-1 structures predicted from DTS and DAA for their binding with a-Syn is presented in Table 5 that are computed using Prodigy as given in step 7 of Section 2.2.2. It showed that the binding energy of interaction with DTS structure model is less than that with DAA, indicating marginally better interaction and stability of binding complex made with DTS structure. Together with the results produced by Table 4 and Figure 4, it shows that the structure predicted through the DTS pipeline is more acceptable than that predicted by DAA.

#### 3.2.2. Result of PMIA Formalism Based Re-Validation of Syn-1 Structure Using Residue Level Interaction

Earlier study based on NMR spectroscopy revealed that the coiled–coil domain (349–557) was responsible for the interaction with the N-terminal region of alpha-synuclein (1–65) [8]. In this regard, to check the correspondence of this result with the Syn-1 and a-Syn, the BIOVIA Discovery studio visualizer 4.5 is utilized as described in step 7 of Section 2.2.2. In this regard, L529, T533, and R536 residues of DAA_synphilin1 coiled–coil domain are found to be involved in interaction with K32, E35, and V40 residues of alpha-synuclein, respectively, as shown in Figure 5, panel C. Furthermore, in the same panel, R536, V537, and K551 residues of DTS_synphilin1 coiled–coil domain interacted with M1, F4, and K12 residues of alpha-synuclein. Direct evidence of involvement of F4 and K12 residues of Synphilin-1 interacting with alpha-synuclein was established previously by NMR study [8]. Therefore, from the above observations, it can be concluded that the DTS_synphilin1 structure is more biologically acceptable compared to DAA_synphilin1.

### 3.3. Result of Structural Details of Interaction Between Best Syn-1 and a-Syn (Both Mutated and Wild Type)

From the result given in the preceding section, it is evident that PMIA formalism helps to find the best structure model of Syn-1 (i.e., the DTS model obtained using AlphaLarge meta-predictor) through the analysis of both binding energy and binding site residues. Furthermore, PMIA formalism is also applied to study the interaction of best Syn-1 structure model with a-Syn (both mutated and wild type) whereas, all these structures were obtained using the steps described in Section 2.2.2. In this second study, PMIA formalism was used both for re-validation of the DAA structure of mutated a-Syn by checking the correspondence with its reported binding characteristics and also to explore new binding information to help in developing new therapeutics. In this regard, Table 6 shows the comparative results of binding energies for the interactions of best Syn-1 structure with both wild and mutated types of a-Syn along with the PDB–RMSD score.

Also, the superimposed structures of wild and mutated a-Syn are shown in panel C of Figure 6 along with the structures of wild and mutated a-Syn in panels A and B, respectively. Visually, although the basic structural characteristics of wild and mutated a-Syn appear not to be very different, the superimposition of them made by PDB application does not appear to represent the same.

The visual representations of the structures of a-Syn, mutated a-Syn, and their alignment are shown in Figure 7 and the results of standard validation for both wild and mutated a-Syn presented in Table 7 show a slight increase in structural disorder in mutated a-Syn in comparison to its wild type.

The number of interactions resolved with the details of the participating residues for the interactions of Syn-1 with a-Syn (wild type) and Syn-1 with a-Syn (mutated) are shown in Table 8. It clearly shows stronger binding for interactions between Syn-1 and mutated a-Syn through a reasonably greater amount of bond formations by the corresponding participating pairs of residues of these proteins in comparison to that between Syn-1 and a-Syn (wild type). It also corresponds well with the binding energy profile of these two interactions as shown in Table 6. Table 8 also shows that several C-terminal residues of a-Syn (mutated) were found to be involved in altered interaction with Syn-1 (shown in bold) in the resulted complex which tallies with the previous FRET based interaction study [66].

The interactions described in Table 8 are also demonstrated in Figure 7 as shown below.

From the result of Table 8 and known information about domain regions of Syn-1, a-Syn (wild type), and a-Syn (mutated), it is evident that the central coiled coil and C-terminal domain regions of Syn-1 are taking part in interactions with a-Syn (residue no. 1 to 18) following the published result [8]. However, in case of an interaction of Syn-1 with mutated a-Syn, the interaction is mostly at the C-terminal region of Syn-1 as already reported [66] with few interactive residues found in the N terminal region.

## 4. Discussion

Extensive studies have been performed on mutations in the genes like SNCA, PRKN1, PRKN2, PINK1, DJ1, and LRRK2 leading to pathophysiological manifestation of Parkinson’s disease (PD) [67,68,69,70,71,72,73]. These mutated genes are known to negatively impact both autophagy and proteosome activities responsible for degradation of misfolded protein aggregates, like Lewy Bodies (LB) [74]. Among these mutations, SNCA-triggered mutations in a-syn, A30P, A53T, and E46K are considered as very important [10,11,12] for its implication in the interaction with Syn-1 leading to formation of these toxic aggregates [3]. Currently, there is no way to stop the progress of this aggregate-formation causing irreversible neuronal damage gradually with time. Under this circumstance, this work focused on the study on the interactions of Syn-1 with mutated a-Syn to extract all the structural details of these individual proteins as well as their bindings that may help in designing target specific therapeutics in future.

To accomplish the aim of this work in unearthing the structural details of interaction of Syn-1 with mutated a-Syn, the information is gradually unfolded through the series of steps as described in the Material and methods and the Result sections that are broadly followed from the logical foundation of Kumar et al. [75]. Consequently, the first attempt in this work was to find highly accurate structures of Syn-1 and mutated a-Syn with specific mutations, A30P, A53T, and E46K, using the semiautomatic pipeline-based meta-predictor, AlphaLarge, as described in Section 2.2.2. The PMIA formalism described in this work basically mines the experimentally valid information relevant to the target structures in question from existing publications and utilizes them to strengthen the choice of the structural models thereafter. There are two sub-pipelines within AlphaLarge named DTS (divide and conquer based strategy giving derived template sources) and DAA (direct application of AlphaFold2). The biological basis of the divide and conquer strategy can be drawn from the existing concept of protein structural folds [76,77]. The fragments are intentionally made to overlap to help the homology application Modeller in assigning the coordinates better by cross-checking at the overlapping zones. Also, each fragment being smaller in size, is found to be better predictable by AlphaFold. It is further confirmed by the observation that the overall MSA-depth found from AlphaFold application for the smaller fragments is deeper than that obtained for the undivided large sequence hinting about the reason for better accuracy under DTS pipeline following [14]. However, this work avoided the pre-detection of the domains since domains are a large structural part; the error in their predictions [51] may have a cascading effect to accumulate a greater amount of error afterwards. In this regard, the potential of AlphaLarge is evident from the standard validation and PMIA formalism-based validation of the output structure models of the training sample proteins as shown in Table 2. All these proteins are of a size similar to the target Syn-1. The re-validation of these structures under PMIA formalism through checking the already-reported interactions in terms of proximity of binding sites and the binding energies of these interactions are shown in Table 3, which clearly demonstrates their potential in selecting a better structure model for a specific protein. It is well accepted [78,79] that not only binding energy, but also the geometric parameters related to the binding sites of protein mediated interactions and bonds are the most important interaction parameters. This is also quite evident since interaction happens through the surface via a lock-and-key type of arrangement whereas, an interaction site that is completely different from the known one may exhibit nearly the same binding energy. This is true for the formation of bonds also that may be similar at different binding sites. That is why the proximity of the binding sites of the model structures outputted by both AlphaLarge and AlphaFold with the ground truth (i.e., the known binding site of the original protein) have been provided in Table 3. Since, for a fixed spatial arrangement and chemical environment of interacting residues (atoms in details), the bond information cannot be altered, in our work, for all protein mediated interactions, the bond information can be considered as embedded within all such interacting residues. For this reason, no separate information on bonds is provided in the manuscript. For details on the binding site residues, please refer to the Appendix A given in the updated Appendix A.

In line with this strategy, to fulfill the main objective to explore the structural aspects of interactions of Syn-1 with a-Syn (mutated), first, the structure of Syn-1 is obtained through its standard validation-based selection from the structure models outputted both by the DTS and DAA sub-pipelines of AlphaLarge as clarified in Table 4 and Figure 4. Continuing with the earlier strategy, the best structure model (DTS) is re-confirmed as the best one for the further study by the application of PMIA based formalism using the results of its interaction with a-syn (wild type) by checking the binding energy as shown in Table 5 and the number of interactions resolved as provided in result Section 3.2.2, the first column of the Table 8, and Figure 5.

Finally, the interaction of Syn-1 with mutated a-Syn is studied following the methodology described in Section 2.2.2. For this purpose, first, the specific three mutations, A30P, A53T, and E46K, reported to promote the formation of aggregates of Syn-1 and a-syn complex, are manually made in the sequence of wild type a-Syn. Finally, the structure of mutated a-Syn is obtained through the DAA pipeline of AlphaLarge. At this step the question was about the authenticity of the structure of the mutated a-Syn. To resolve it, the same earlier validations, both standard and PMIA formalism are applied. The result of standard validation, as given in Figure 6 and Table 7, shows a slight increase in structural disorder in mutated a-Syn compared to its wild counterpart. At this stage, PMIA formalism helps establish the authenticity of the mutated a-Syn structure and explores the structural basis of interactions between Syn-1 and mutated a-Syn.

Since mutated a-Syn is already known to promote formation of higher amount of aggregates of Syn-1 and a-Syn complex in comparison to that made by a-Syn (wild type), it was intriguing to see whether it is reflected from the binding energy profiles of protein complexes made by the interactions of Syn-1 with a-Syn and mutated a-Syn, respectively. In this regard, the result of binding energy profile as shown in Table 6 clearly shows reasonably stronger binding of Syn-1 with mutated a-Syn in comparison to that with a-Syn (wild type). Furthermore, it can also be expected that for stronger interactions of Syn-1 with mutated a-Syn, the number of residue pairs of both the proteins participating in this interaction should be more comparable to the interaction of Syn-1 with a-Syn (wild type). Along with the visual demonstration of this interaction in Figure 7, it is also intriguing to observe that the same is also satisfied by the results shown in Table 8. The result of Table 8 is found to give new insight into the detailed residue–residue pair wise interactions between syn-1 and mutated a-Syn. In this regard, several C-terminal residues of a-Syn (mutated) were found to be involved in altered interaction with Syn-1 (shown in bold in Table 8) in the resulted complex which tallies with the previous FRET based interaction study [65]. Therefore, in this work, the PMIA formalism backed by AlphaLarge appears to pave the way for the new therapeutics to stop the progression of PD by offering the new structural insight of the causal interaction of Syn-1 with mutated a-Syn.

## 5. Conclusions

Popular therapeutics mainly rely on protein level control targeting the causal proteins and their interactions leading to the particular pathophysiological process. However, to fulfill this goal, there must be the availability of the structures of these proteins where experimental methods to get them are too slow and the prediction methods are too error-prone until the launch of AlphaFold2 by Google Colab that has already gained popularity for its prediction accuracy. Consequently, this work focused on study of interactions of the proteins Syn-1 and a-Syn with specific mutations leading to their aggregates referred to as Lewy Bodies (LB) that are toxic enough to cause irreversible neuronal damage causing PD. In line with this objective, a semiautomatic pipeline-based meta-predictor, AlphaLarge, compensates for the limitation of AlphaFold2 in predicting the structure of large proteins like Syn-1. The interaction of Syn-1 with a-Syn is studied under PMIA formalism, which not only re-validated the structures of these proteins, but also revealed important structural details of their interaction in terms of the specific residue–residue pairs participating in this interaction. This crucial information is expected to offer better therapeutic strategy in future to control the progression of PD.

## Figures and Tables

**Figure 1 biomolecules-14-01588-f001:**
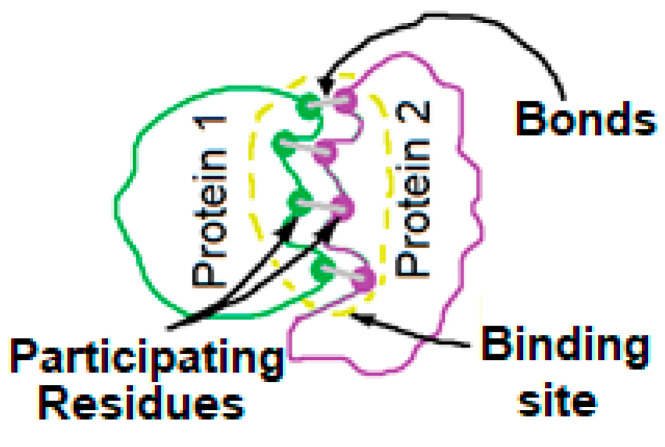
Working parameters (participating residues, bonds, and binding energies) used in PMIA.

**Figure 2 biomolecules-14-01588-f002:**
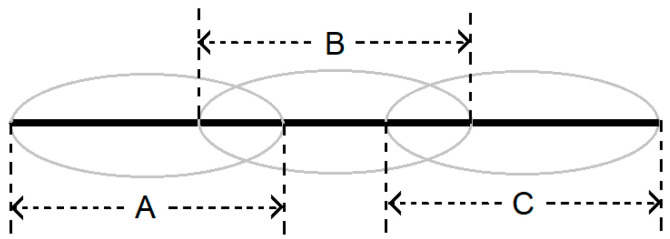
Creation of overlapping fragments A, B, and C following the divide and conquer strategy.

**Figure 3 biomolecules-14-01588-f003:**
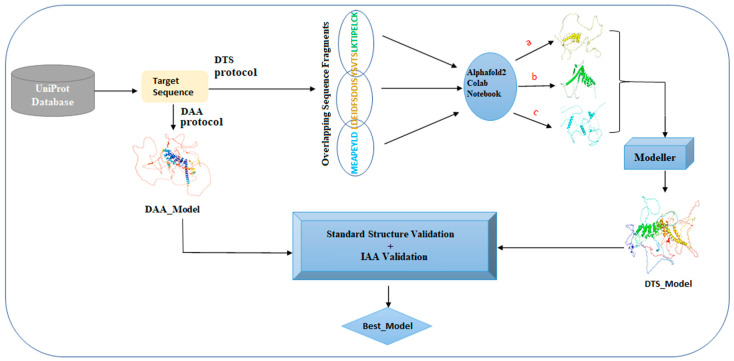
Flowchart depicting the working principle of AlphaLarge, where a, b and c shows the structure of fragments 1, 2 and 3 respectively.

**Figure 4 biomolecules-14-01588-f004:**
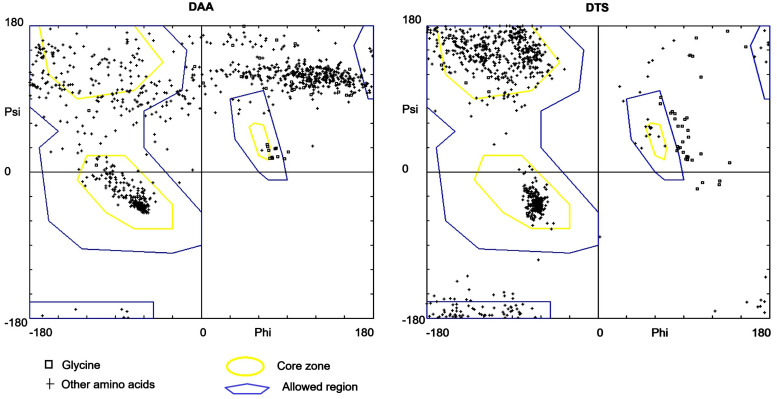
Quality assessment of Syn-1 structure predicted by DAA and DTS method from Ramachandran plots obtained from the Swiss-PdbViewer.

**Figure 5 biomolecules-14-01588-f005:**
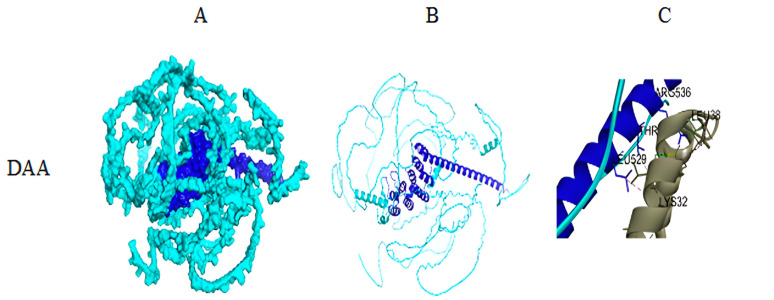
(**A**) Surface filling model, (**B**) ribbon representation of DTS and DAA structures of Syn-1 (cyan overall with deep blue highlighting central coiled–coil region), and (**C**) zoomed version of (**B**) showing interactions with a-syn (wild type) (grey ribbon).

**Figure 6 biomolecules-14-01588-f006:**
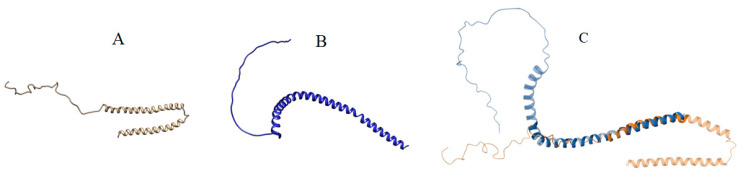
Panels (**A**–**C**) show wild, mutated, and superimposed structures of wild (PDB ID: 1XQ8), and mutated a-Syn, respectively.

**Figure 7 biomolecules-14-01588-f007:**
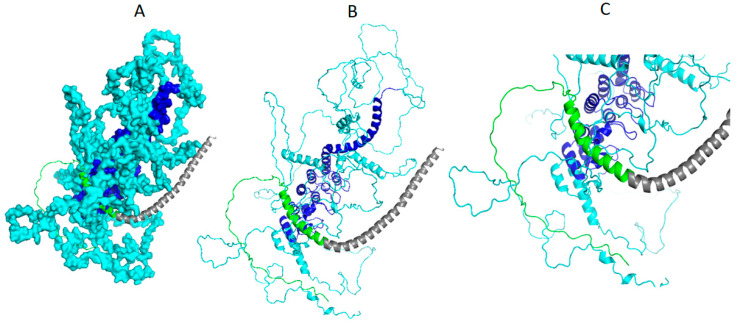
Syn-1 and a-Syn (mutated) complex structure: (**A**) Surface filling model of the AlphaLarge predicted structure of Syn-1 (cyan overall with deep blue highlighting central coiled coil region) and ribbon representation of mutated a-syn (N-terminal 1–65 in grey ribbon), (**B**) ribbon representation of the complex and (**C**) zoomed version of (**B**) showing interactions with mutated a-syn through C-terminal (green ribbon).

**Table 1 biomolecules-14-01588-t001:** Training proteins and their corresponding ligands.

Proteins (PDB ID)	Ligands (Chemical ID)
Mutant monomer of recombinant human hexokinase Type I complexed with Glucose, Glucose-6-Phosphate, and ADP (1CZA)	alpha-D-glucose 6-phosphate(G6P) [26]
Crystal structure of the aldehyde oxidoreductase from desulfovibriodesulfuricansatcc 27774(1DGJ)	Pterin cytosine dinucleotide(MCN) [27,28,29]
Crystal structure of human leucyl-tRNA synthetase, Leu-AMS-bound form(6KIE)	5′-O-(L-leucylsulfamoyl)adenosine(LSS) [30,31]
Structure determinants of phosphoinositide 3-kinase inhibition by wortmannin, LY294002, quercetin, myricetin and staurosporine(1E7U)	Wortmannin (KWT) [32,33]
X-ray crystal structure of human ceruloplasmin at 3.0 angstroms(1KCW)	2-acetamido-2-deoxy-beta-D-glucopyranose(NAG) [34]
Crystal structure of endoplasmic reticulum aminopeptidase 2 (erap2) complex with a highly selective and potent small molecule(7SH0)	(2S)-N-hydroxy-3-(4-methoxyphenyl)-2-[4-({[5-(pyridin-2-yl)thiophene-2-sulfonyl]amino}methyl)-1H-1,2,3-triazol-1-yl]propenamide (GIY) [35]
OmecamtivMercarbil binding site on the Human Beta-Cardiac Myosin Motor Domain(4PA0)	methyl 4-(2-fluoro-3-{[(6-methylpyridin-3-yl)carbamoyl]amino}benzyl)piperazine-1-carboxylate(2OW) [36,37]
Structure of human DNMT1 (601-1600) in complex with Sinefungin(3SWR)	SINEFUNGIN(SFG) [38]
Structure of the m1 alanylaminopeptidase from malaria complexed with a hydroxamic acid-based inhibitor(4R5X)	3-amino-N-{(1R)-2-(hydroxyamino)-2-oxo-1-[4-(1H-pyrazol-1-yl)phenyl]ethyl}benzamide(R5X)[39,40]
Structure of Ca2+ ATPase(5ZTF)	Phosphomethylphosphonic acid adenylate ester(ACP) [41,42]
Structure of C-terminal fragment of Vip3A toxin(6VLS)	DI(HYDROXYETHYL)ETHER(PEG) [43,44]
E.coli beta-galactosidase (E537Q) in complex with fluorescent probe KSA02(7BRS)	8-[2-[(E)-2-[4-[(2S,3R,4S,5R,6R)-6-(hydroxymethyl)-3,4,5-tris(oxidanyl)oxan-2-yl]oxyphenyl]ethenyl]-3,3-dimethyl-indol-1-ium-1-yl]octanoic acid(F4X) [45,46]
Crystal structure of human MTR4(6IEG)	ADENOSINE-5′-DIPHOSPHATE(ADP) [47]
Structure of human sodium-calcium exchanger NCX1(8JP0)	2-{4-[(2,5-difluorophenyl)methoxy]phenoxy}-5-ethoxyaniline(EKY) [48]

**Table 2 biomolecules-14-01588-t002:** A profile of W_DAA_, W_DTS_, and Tie percentage against various structure validation parameters.

Standard Structure Validation Metrics	$W_{DTS}$	$W_{DAA}$	Tie
Ramachandran Score	100	0	0
G-factor	93	7	0
GDT–TS	7	36	57
PAL_RMSD	43	28	29
QMEAN	50	43	7
SRI	43	43	14
ERRAT	7	93	0
TM-Score	7	43	50

**Table 3 biomolecules-14-01588-t003:** The comparative binding profiles of interactions obtained for individual proteins using experimental structure, DTS and DAA models, respectively. Better performance of DTS over DAA is highlighted by bold font.

Proteins	Ligands	Interactions Details				
		Proximity of Binding Site of Experimental Structure in Å with That of		Binding Energy in kJ/mol		
		DAA	DTS	BI_Experimental_	BI_DAA_	BI_DTS_
**1CZA**	G6P	**33.3**	**27.8**	−25.9408	−28.0328	−27.6144
1DGJ	MCN	**29.2**	**28.6**	−59.8312	−10.8784	−24.2672
6KIE	LSS	**13.0**	**9.0**	−19.2464	−23.4304	−12.9704
1E7U	KWT	**28.9**	**24.3**	−23.4304	−15.4808	−15.4808
1KCW	NAG	11.6	16.5	−17.9912	−22.1752	−22.5936
7SH0	GIY	**11.8**	**6.2**	−33.0536	−27.196	−26.7776
4PA0	2OW	11.9	15.9	−30.1248	−23.4304	−25.9408
3SWR	SFG	11.2	18.5	−40.1664	−22.1752	−18.4096
4R5X	R5X	22.9	36.5	−22.5936	−17.1544	−16.3176
5ZTF	ACP	**11.6**	**7.7**	−30.5432	−17.5728	−26.7776
6VLS	PEG	**31.3**	**10.1**	−1.6736	−0.4184	−6.6944
7BRS	F4X	**21.4**	**4.7**	−22.5936	−25.5224	−20.0832
6IEG	ADP	13.9	22.6	−21.3384	−23.8488	−15.0624
8JP0	EKY	**16.8**	**10.5**	−32.6352	−22.5936	−10.46

**Table 4 biomolecules-14-01588-t004:** Comparison of Syn-1 model structures predicted through DTS and DAA pipelines using the validation parameters, ERRAT, Ramachandran score, G-factor, QMEAN Score, and % coil amount present in the structures.

Models	Ramachandran Score in % for Different Regions				G-Factor	QMEAN Score	% Coil	ERRAT
	Core	Allowed	Generously Allowed	Disallowed				
**DTS**	**82.1**	**14.9**	**2.1**	0.9	−0.15	−5.97	65.6	72.48
**DAA**	32.2	17	18.3	32.5	−2.55	−21.82	72.7	60.27

**Table 5 biomolecules-14-01588-t005:** Binding energy (BI) for interaction of a-syn with Syn-1 (both DTS and DAA models).

Structure Model of a-Syn	Structure Models of Syn-1	BI in kJ/mol
Wild type (PDB ID: 1XQ8)	**DTS**	−58.576
	**DAA**	−56.0656

**Table 6 biomolecules-14-01588-t006:** BI for interaction of Syn-1 best model (DTS) with both wild and mutated a-Syn along with the Backbone PDB–RMSD between wild and mutated a-Syn in Å (R).

Structure Model of Syn-1	Structure Models of a-Syn	BI in kJ/mol	PDB–RMSD (R) in Å with 11% Identity
DTS	**Wild type (PDB ID: 1XQ8)**	−58.576	3.35
	**Mutated a-Syn**	−110.0392	

**Table 7 biomolecules-14-01588-t007:** Comparison of structures of wild and mutated a-Syn using the validation parameters, ERRAT, Ramachandran score, G-factor, QMEAN Score, and % coil amount present in the structures.

Models	Ramachandran Score in % for Different Regions				G-Factor	QMEAN Score	% Coil	ERRAT
	Core	Allowed	Generously Allowed	Disallowed				
**Wild Type (PDB ID: 1XQ8)**	**77.4**	**13**	**7.8**	1.7	−0.05	−5.59	25	73.01
**Mutated** **a-Syn**	67.5	14.9	11.4	6.1	−1.95	−11.25	34.2	0

**Table 8 biomolecules-14-01588-t008:** A comparative profile of residue level interactions of Syn-1 (A) with a-syn (wild type) (B) and Syn-1 (A) with a-syn (mutated) (C) (the reported interactions [8,66] are shown in bold).

Interaction of Syn-1 with a-Syn (Wild Type)	Interaction of Syn-1 with a-syn (Mutated)
A: LEU547—B: LEU8 A: LYS551—B: MET1 A: LYS626—B: VAL3 A: ILE619—B: VAL3 A: LEU623—B: VAL3 A: **VAL537—B: LYS12** A: VAL816—B: ALA18 A: **ARG536—B: PHE4** A: GLU627—B: MET1 A: GLU630—B: MET1	A:PRO699—C: ALA11 A: ARG700—C: ALA11 A: ALA696—C: ALA18 A: ARG899—C: LYS58 A: PRO907—C: GLY73 A: **GLY52—C: VAL118** A: ARG700—C: GLY7 A: ARG899—C: THR54 A: THR901—C: VAL55 **A: SER19—C: TYR133** **A: ARG34—C: TYR125** **A: ARG34—C: GLU126** **A: CYS51—C: ASN122** A: TYR158—C: THR81 A: LYS160—C: LYS80 A: SER200—C:ASP98 A: ARG700—C: GLY7 A: ARG899—C: THR54 A: THR901—C: VAL55 **A: SER21—C: GLU130** **A: ARG33—C: TYR125** **A: TRP49—C: ALA124** **A: GLN307—C: TYR133** A: SER912—C: THR75 A: GLN164—C: THR92 A: LEU165—C: THR92 A: GLN164—C: GLY93 **A: ASP36—C: SER129** **A: SER21—C: GLN134**

## Data Availability

All the data used in this work are available from the authors and can be shared on such demand.

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
