# Peer review of "A Structural Proteomics Exploration of Synphilin-1 and Alpha-Synuclein Interaction in Pathogenesis of Parkinson’s Disease"

_biomolecules, 2024, doi:10.3390/biom14121588_

Round 1
Reviewer 1 Report
Comments and Suggestions for Authors
In this manuscript, the authors have created a semi-automatic pipeline based on AlphaLarge to predict high-fidelity structures of large proteins like synphilin-1.
The structure models were validated by a deep-learning program in order to study the interactions between both molecules.
The manuscript is well written, interesting and is easy to read.
I would just like to point out a few points from the paper:
In the introduction it is stated that there are 3 main causes leading to amyloid aggregation of a-Syn, although the 3 causes given would not be the main causes, only the first of the causes, a SNCA mutation leading to over-expression of a-Syn. Pathological mutations in the a-Syn molecule would be another one, and there are quite a few more main causes before the interaction with Syn-1. Perhaps that point should be rewritten to simply say that among the main causes that lead the amyloid aggregation of a-Syn is interaction with Syn-1n at pathological mutation points.
The authors use a certain structure of a-Syn, despite the fact that it is an intrinsically disordered protein, without justifying why they consider that in the process of interaction between Syn-1 and a-Syn the latter will present the structures that a-Syn can present when it interacts with lipid vesicles. I understand that since these structures of a-Syn are available in the PDB, they are the ones used by AlphaFold (which works by deep learning) to predict the structure of a-Syn. But perhaps it would be necessary to justify why they have chosen these structures or at least explain why they have decided to accept that an intrinsically disordered protein acquires a certain structure in its interaction with another protein.
Overall, I find the paper very interesting and I think I have nothing more to contribute than my congratulations to the authors for the present work.
Author Response
Comment 1
In this manuscript, the authors have created a semi-automatic pipeline based on AlphaLarge to predict high-fidelity structures of large proteins like synphilin-1.
The structure models were validated by a deep-learning program in order to study the interactions between both molecules.
The manuscript is well written, interesting and is easy to read.
I would just like to point out a few points from the paper:
Response 1
Thanks to the reviewer for summarizing our work. However, we would like to point out that the structure models in this work were validated by using both the standard validation parameters and the novel PMIA based methods developed in our work to compensate for the experimental validation.
Comment 2
In the introduction it is stated that there are 3 main causes leading to amyloid aggregation of a-Syn, although the 3 causes given would not be the main causes, only the first of the causes, a SNCA mutation leading to over-expression of a-Syn. Pathological mutations in the a-Syn molecule would be another one, and there are quite a few more main causes before the interaction with Syn-1. Perhaps that point should be rewritten to simply say that among the main causes that lead the amyloid aggregation of a-Syn is interaction with Syn-1n at pathological mutation points.
Response 2
Thanks to the reviewer for pointing out the need for converging to the specific objectives. Accordingly, we have rewritten the introduction (page 2, lines 58 to 61) highlighting the changes in red coloured font.
Comment 3
The authors use a certain structure of a-Syn, despite the fact that it is an intrinsically disordered protein, without justifying why they consider that in the process of interaction between Syn-1 and a-Syn the latter will present the structures that a-Syn can present when it interacts with lipid vesicles. I understand that since these structures of a-Syn are available in the PDB, they are the ones used by AlphaFold (which works by deep learning) to predict the structure of a-Syn. But perhaps it would be necessary to justify why they have chosen these structures or at least explain why they have decided to accept that an intrinsically disordered protein acquires a certain structure in its interaction with another protein.
Response 3
Thanks to the reviewer for raising this issue of interaction of intrinsically disordered proteins (IDPs). In this regard it can be said that IDPs are more flexible than normal proteins in regards to their ability to modulate their internal structure. Alpha-synuclein, an intrinsically disordered protein, undergoes structural transitions under specific conditions, including a notable shift from an alpha-helix to a beta-sheet conformation. In its interaction with lipid membranes (PDB ID 1XQ8) and proteins like calmodulin (PDB ID: 2M55), alpha-synuclein adopts a stable alpha-helical structure. Consequently, for studying its interaction with Synphilin-1, the alpha-helical conformation of full-length wild-type alpha-synuclein (PDB ID:1XQ8) and mutated alpha-synuclein, representing a minimum-energy state as predicted by AlphaFold2, were selected as optimal modelling. Accordingly, we have rewritten the revised manuscript (page 10, lines 329 to 334) highlighting the changes in red coloured font.
Comment 4
Overall, I find the paper very interesting and I think I have nothing more to contribute than my congratulations to the authors for the present work.
Response 4
Thanks to the reviewer for the complements.
Reviewer 2 Report
Comments and Suggestions for Authors
Tripathi et al. mainly explored how the interaction between Synphilin-1 and mutant α-synuclein (α-syn) contributes to the pathogenesis of Parkinson's disease (PD). The focus of the study was to use the newly developed semi-automated structure prediction method AlphaLarge to identify specific structural binding sites and interaction mechanisms to gain a deeper understanding of this pathological process. Through a "divide and conquer" strategy, proteins with larger molecular weights were reasonably divided, which to a certain extent overcame the limitations of AlphaFold in the prediction of large multi-domain proteins. In addition, combined with the PMIA framework, the reliability and biological relevance of the predicted structure were demonstrated by combining multiple verification indicators such as energy, binding sites, and interacting residues.
Of course, although the authors also listed several other examples of comparison results between predictions and experimental data, unfortunately, there is no specific experimental evidence. However, the authors made up for this defect to a certain extent through the quantitative analysis of Ramachandran scores. Moreover, if the quantitative analysis results of Table 2 and Figure 4 can be linked, a more detailed explanation of the confidence given from different angles of different indicators will be more conducive to reflecting the advantages of the AlphaLarge method in structure prediction. In addition to AlphaFold, adding a comparison with the Rosetta prediction method will also highlight the advantages of the strategy better.
Author Response
Comment 1
Tripathi et al. mainly explored how the interaction between Synphilin-1 and mutant α-synuclein (α-syn) contributes to the pathogenesis of Parkinson's disease (PD). The focus of the study was to use the newly developed semi-automated structure prediction method AlphaLarge to identify specific structural binding sites and interaction mechanisms to gain a deeper understanding of this pathological process. Through a "divide and conquer" strategy, proteins with larger molecular weights were reasonably divided, which to a certain extent overcame the limitations of AlphaFold in the prediction of large multi-domain proteins. In addition, combined with the PMIA framework, the reliability and biological relevance of the predicted structure were demonstrated by combining multiple verification indicators such as energy, binding sites, and interacting residues.
Response 1
Thanks to the reviewer for providing a magnificent summary of our work.
Comment 2
Of course, although the authors also listed several other examples of comparison results between predictions and experimental data, unfortunately, there is no specific experimental evidence. However, the authors made up for this defect to a certain extent through the quantitative analysis of Ramachandran scores.
Response 2
We agree with the reviewer that we have made up for experimental validation in our work. However, we did it through our novel validation method based on a Protein Mediated Interaction Analysis (PMIA) that uses already published experimental reports on the interaction of the target protein with other proteins or ligands. Since for most of the proteins without known structures, at least information of their interactions with their interacting partners (other proteins or ligands) are available, PMIA method can well be applied to validate their structures.
Comment 3
Moreover, if the quantitative analysis results of Table 2 and Figure 4 can be linked, a more detailed explanation of the confidence given from different angles of different indicators will be more conducive to reflecting the advantages of the AlphaLarge method in structure prediction.
Response 3
We think the reviewer meant the link between table 4 and Figure 4 instead of Table 2 and Figure 4. Accordingly, we have made changes in the manuscript in line 310 to 319 (from page 9 to 10).
Comment 4
In addition to AlphaFold, adding a comparison with the Rosetta prediction method will also highlight the advantages of the strategy better.
Response 4
Honouring the reviewer’s concern, we have applied RoseTTAFold method in our divide and conquer protocol on syn-1 that resulted in the structure we refer as Ros. Ros showed quite similar outputs for standard validation parameters in comparison to DTS. However, after the application of PMIA method it shows substantially poor result in comparison to that for DTS structure. We present the result in the following Table:
Table: Results of PMIA validation for Ros and DTS structures of syn-1 through its interaction with wild type and mutated a-syn.
a-syn type |
Binding energy in KJ/mol |
|
Ros |
DTS |
|
Wild |
-33.4 |
-58.6 |
Mutated |
-26.9 |
-110.0 |
Also, currently, RoseTTAFold server has a sequence size limit of 700 for which direct application of RoseTTAFold for large proteins is not possible. Therefore, we have not proceeded to apply RoseTTAFold in our work.
Reviewer 3 Report
Comments and Suggestions for Authors
The prediction of interactions between disordered proteins is a challenging work, especially for large disordered proteins. Both AlphaFold and AlphaFold2 designed by Google have certain limitations in predicting large protein structures. In this work, the authors designed AlphaLarge, which greatly improved the prediction accuracy compared to directly applying AlphaFold2. The authors explored the detailed structural aspects of the interaction between α-synuclein and synphilin-1 to identify specific binding sites using AlphaLarge. Here are some concerns that need to be addressed.
1. In this manuscript, the authors predicted the interactions between synphilin-1 and WT α-synuclein, α-synuclein A30P, E46K, and A53T mutations. How about other disease mutations of α-synuclein, such as H50Q, G51D and A53E?
2. Is AlphaLarge universal in predicting the structure of large proteins? Another large protein structure prediction by AlphaLarge is recommended.
3. The resolution of the figures needs to be improved, especially the Figure 4.
Author Response
The prediction of interactions between disordered proteins is a challenging work, especially for large disordered proteins. Both AlphaFold and AlphaFold2 designed by Google have certain limitations in predicting large protein structures. In this work, the authors designed AlphaLarge, which greatly improved the prediction accuracy compared to directly applying AlphaFold2. The authors explored the detailed structural aspects of the interaction between α-synuclein and synphilin-1 to identify specific binding sites using AlphaLarge. Here are some concerns that need to be addressed.
Comment 1
In this manuscript, the authors predicted the interactions between synphilin-1 and WT α-synuclein, α-synuclein A30P, E46K, and A53T mutations. How about other disease mutations of α-synuclein, such as H50Q, G51D and A53E?
Response 1
Thanks to the reviewer for pointing out about more possible mutations and their effects in the interactions of a-syn with syn-1. In this regard, Fevga C et al (2021) [Citation: A new alpha-synuclein missense variant (Thr72Met) in two Turkish families with Parkinson's disease. Parkinsonism Relat Disord. 89:63-72. doi: 10.1016/j.parkreldis.2021.06.023] pointed out about existence of 19 possible mutations across various racial types among which A53T is predominant. Furthermore, some of these mutations are responsible for early development of amyloid aggregates (e.g., A53T) while some are responsible for late formation (e.g., H50Q). Moreover, simultaneous occurrences of A53T, A30P and E46K are reported in earlier work as referred in our manuscript, whereas, other combinations of mutations are yet to be studied. For this reason, in this work, we have focused our study to examine the effect of specific combination of mutations.
Comment 2
Is AlphaLarge universal in predicting the structure of large proteins? Another large protein structure prediction by AlphaLarge is recommended.
Response 2
In this regard, we would like to draw the kind attention of the reviewer to the result of benchmarking with 14 large proteins as shown in Table 3. Table 3 indicates that among 14 proteins, DTS outperformed DAA for 9 proteins in regards to the proximity of binding site of the model structure with that of the experimental structure.
Comment 3
The resolution of the figures needs to be improved, especially the Figure 4.
Response 3
Honouring the reviewer’s concern we have replaced the UCLA SAVES figure with that obtained from MolProbity server for the same structures and incorporated the necessary changes in the caption of the Figure 4 (Line 322 of page 10).
Round 2
Reviewer 3 Report
Comments and Suggestions for Authors
The authors addressed my concern.